# Improving Nutrition Care, Delivery, and Intakes Among Hospitalised Patients: A Mixed Methods, Integrated Knowledge Translation Study

**DOI:** 10.3390/nu11061417

**Published:** 2019-06-24

**Authors:** Shelley Roberts, Lauren T. Williams, Ishtar Sladdin, Heidi Neil, Zane Hopper, Julie Jenkins, Alan Spencer, Andrea P. Marshall

**Affiliations:** 1School of Allied Health Sciences, Griffith University; Gold Coast Campus, Southport QLD 4222, Australia; lauren.williams@griffith.edu.au (L.T.W.); i.sladdin@griffith.edu.au (I.S.); heidi.neil@griffithuni.edu.au (H.N.); 2Menzies Health Institute Queensland, Griffith University; Gold Coast Campus, Southport QLD 4222, Australia; a.marshall@griffith.edu.au; 3Gold Coast Hospital and Health Service; 1 Hospital Blvd, Southport QLD 4219, Australia; zane.hopper@health.qld.gov.au (Z.H.); julie.jenkins@health.qld.gov.au (J.J.); alan.spencer@health.qld.gov.au (A.S.); 4School of Nursing and Midwifery, Griffith University; Gold Coast Campus, Southport QLD 4222, Australia

**Keywords:** clinical nutrition, complex interventions, hospitalised patients, knowledge translation, malnutrition, nutrition care, research co-development

## Abstract

Malnutrition is a common and complex problem in hospitals. This study used an integrated knowledge translation approach to develop, implement, and evaluate a multifaceted, tailored intervention to improve nutrition care, delivery, and intake among acute medical inpatients. This observational, pre-post study was conducted in a medical ward at a public hospital in Australia. The intervention was co-developed with key stakeholders and targeted three levels: individuals (nutrition intake magnets at patient bedsides), the ward (multidisciplinary hospital staff training), and the organisation (foodservice system changes). Observational data were collected pre- and post-intervention on patient demographics, food intakes, and the mealtime environment. Data were entered into SPSS and analysed using descriptive and inferential statistics. Ethical approval was gained through the hospital and university ethics committees. A total of 207 patients were observed; 116 pre- and 91 post-intervention. After intervention implementation, patients’ mean energy and protein intakes (in proportion to their estimated requirements) were significantly higher and the number of patients eating adequately doubled (*p* < 0.05). In summary, a multifaceted, pragmatic intervention, tailored to the study context and developed and implemented alongside hospital staff and patients, seemed to be effective in improving nutrition practices and patient nutrition intakes on an acute medical ward.

## 1. Introduction

Malnutrition is a serious clinical issue, affecting between 20–50% of hospitalised patients in Australia and worldwide [1,2]. Malnutrition increases risks of mortality and complications and results in longer hospital stays, more frequent readmissions and increased costs, placing a large burden on patients and the health care system [3,4]. Along with disease, inadequate oral intake is the primary cause of malnutrition in the clinical setting [5]. Many studies have shown that patients eat inadequately to meet their requirements in hospital, placing them at nutritional risk [6,7,8,9]. A complex mix of factors affect the dietary intake of patients, including those relating to the patient themselves, the hospital environment and foodservice, and nutrition care received [10].

Over the past few decades, numerous interventions have been developed and tested to improve the food intake of patients and combat malnutrition in hospitals. However, initiatives such as protected meal times [11], feeding assistance [12,13], dietary education and counselling [14], and communal dining [15] have shown modest or limited success. Meanwhile, interventions that have been shown to increase dietary intake, such as providing blanket oral nutrition supplements to all patients [16], have not been routinely adopted into practice. The inability to make substantial and sustainable changes to the nutritional status of hospitalalised patients is likely due to the number and complexity of issues contributing to malnutrition in this setting. Specific issues and barriers to adequate nutrition are likely to vary between different settings and populations; so these must be assessed within local contexts and intervention strategies selected, tailored, implemented, and tested accordingly.

One approach to designing and implementing interventions that are likely to be effective and sustainable in practice is a knowledge translation (KT) approach. KT research focuses on the ‘synthesis, dissemination, exchange and ethically sound application of knowledge’ to improve health and health services [17]. In particular, integrated KT has a strong focus on context and involves targeted stakeholder engagement in all phases of research, including problem assessment, intervention development and evaluation, and dissemination and implementation of research findings [17]. This approach ensures interventions address problems specific to the context and are relevant, acceptable, and feasible for end-users. One study using a KT approach (PARIHS framework) to develop and implement an intervention to facilitate use of malnutrition screening tools and innovation in nutritional care found this was effective in changing nutrition care and practice in the hospital setting [18].

This study aimed to use an integrated KT approach to develop, implement, and evaluate an intervention to improve nutrition care, delivery, and intake among acute medical inpatients.

## 2. Methods

### 2.1. Study Overview

This observational, pre-post study was guided by the four-step approach described by French et al. [19] along with the Medical Research Council (MRC) guidance for developing and evaluating complex interventions [20]. French’s fours step approach includes: identifying who needs to do what differently, identifying barriers and enablers that need to be addressed using a theoretical framework, deciding on strategies appropriate to enhance enablers and overcome barriers, and finally interpret the behaviour change that occurs [19]. Intervention development and implementation were underpinned by patient-centred care as the conceptual framework [21] and guided by the Knowledge to Action cycle [17]. Ethical approval was gained through the participating health service (HREC/14/QHC/7) and university (NRS/22/14/HREC), which included approval for waived consent from participants. Institutional approval to access hospital data was obtained. The study was guided by a Nutrition Reference Group comprising relevant academic and hospital staff.

### 2.2. Setting and Participants

This study was conducted in a 28-bed Acute Medical Unit (AMU) at a public metropolitan teaching hospital in southeast Queensland, Australia. Patients were included in the study if they were: (a) ≥18 years of age, (b) able to take food or fluids orally and (c) able to communicate in English (verbally and in writing). Patients were excluded if they were: (a) not expected to survive the 48 hours from recruitment or were not eligible for full aggressive care; or (b) admitted with a diagnosis or history of an eating disorder. Patients admitted to the ward between February–March 2015 (pre intervention) and March–April 2016 (post intervention) were consecutively recruited and followed until discharge or until the end of the data collection period, whichever occurred first.

### 2.3. Study Preparation and Pilot Testing

To maximise the accuracy of patient food intake calculations, a foodservice audit was conducted to ensure serving sizes provided by the hospital kitchen accurately reflected those specified by the hospital’s electronic foodservice system (from which nutritional data were sourced). This audit resulted in a change in the serving utensils used by food service staff to improve serving size standardisation. Five research assistants were involved in the study and were trained in the use of the observational data collection tool and study methods.

### 2.4. Intervention

The intervention was complex and multifaceted and targeted individual, ward, and organisational levels. Intervention strategies were developed in conjunction with key stakeholders and informed by earlier phases of research, including assessment of the problem and barriers within the local context [22,23]. At the individual level, nutrition intake magnets (using a traffic light system) were displayed on each patients’ whiteboard to flag nutritional risk. At the ward level, discipline-specific training was conducted with nurses, doctors, and foodservice staff, tailored to each group based on findings around barriers and facilitators to optimal nutrition from earlier phases. At an organisational level, foodservice system changes were introduced, including the addition of a new ‘Full + Hot Breakfast’ (Full + Hot BF) diet code for patients (which included the addition of two different hot breakfast items each day to the hospital’s regular diet) and moving the breakfast meal to around 30–60 minutes earlier in the morning to facilitate meals being provided before ward rounds and/or transfer of patients for tests or procedures. Intervention strategies were implemented by internal facilitators who were clinicians and members of the study team.

### 2.5. Data Collection

Two rounds of observational data collection were undertaken; pre and post intervention. Patient data, including demographic (age, gender); medical (diagnosis, comorbidities, length of stay); and nutritional (height, weight, BMI, dietitian input, prescribed supplements) data were collected. For each round, a data collection schedule was developed, ensuring that every weekday of the two-week cyclic menu was observed over 10 days of data collection. Patients were observed by research assistants from 7:00 a.m.–7:00 p.m. hours each day to include all main and mid meals in the collection. The mealtime environment was observed before and during meals to assess food service and nutrition care delivery. As each meal occasion was treated as its own event, patients were observed more than once over their hospital stay to collect data on the mealtime environment, such as placement of meal trays, set-up, and feeding assistance received and any mealtime interruptions. Patients’ plates were observed at the end of each main meal (breakfast, lunch, dinner) and the amount of each food item consumed (none, ¼, ½, ¾, all) was recorded using visual estimation, a valid and reliable method for collecting dietary intake data [24]. Patients’ intakes were observed for their entire hospital stay, but nutritional adequacy was calculated for their first day of complete nutrition intake data (i.e., no patients had repeated intakes in the analysis). Some patients were discharged before a full day of nutrition intake could be observed; these patients were excluded from dietary adequacy analyses. If patients consumed <50% of their meal, reasons why were noted by observing and/or asking the patient. Mid-meal intakes (morning tea, afternoon tea, supper) were obtained by observing package waste or by asking the patient. Items brought in by family members or purchased from the hospital cafeteria or vending machines were included in the data collection.

### 2.6. Data Analysis

Amounts consumed of each meal component, food or fluid item, or nutritional supplement provided by the hospital foodservice was entered into the hospital’s electronic foodservice system database (Delegate Technology, Vienna, Austria). Foods consumed from outside the hospital foodservice were analysed using Foodworks^®^ (version 7, AUSNUT 2011-13 database; Xyris Software, Brisbane, Australia). Analyses from both databases were combined to calculate energy and protein intakes. Institutional clinical practice guidelines were used to calculate individual, disease-specific estimated energy requirements (EER) and estimated protein requirements (EPR) [25]. These represent the predicted average daily dietary energy/protein intakes required to maintain energy balance and meet protein needs of individuals of a defined age, gender, weight, and clinical condition, using static equations [25]. Protein and energy intakes were divided by the estimated requirements of patients to calculate what proportion of estimated requirements were met. Adequate energy intake was defined as patients meeting ≥75% of their EER, based on previous research demonstrating this level of intake is sufficient for weight maintenance among hospitalised patients [26]. The same threshold was used for protein.

Data were entered into SPSS version 22 (IBM, Chicago, IL, USA) and 10% of the data was checked for accuracy (yielding <1% errors). Demographic and observational data were analysed descriptively and presented as percentages or as mean ± SD with non-normally distributed data reported as median and interquartile range (IQR). Independent *t*-tests and chi-square tests were used to test for differences in intakes pre- and post-intervention, and for associations between intakes and observed variables. Mann–Whitney U was used for variables not normally distributed. Significance was set at *p* < 0.05 for all associations.

## 3. Results

### 3.1. Demographics

A total of 207 patients were observed; 116 pre and 91 post intervention implementation. Patient demographics are described in Table 1, which shows no differences between the two groups. The most frequently reported diagnoses in both groups were functional/musculoskeletal disorders (26%), respiratory conditions (22%), and infection (20%).

### 3.2. Nutritional Intakes

A full day’s intake (from 7:00 a.m. to 7:00 p.m.) was recorded for 66 (57%) patients pre-implementation and 61 (67%) patients post-implementation; these data were included in nutritional analyses. Some patients were unable to be observed for a full day due to early discharge or being otherwise unavailable for observation. There were no differences (*p* > 0.05) in patient characteristics between those who were and were not included in dietary intake analyses. Patients’ energy and protein intakes, estimated requirements, and proportion of requirements met are shown in Table 2. Patients’ mean proportion of estimated protein requirements met and the number of patients consuming adequate energy and protein (i.e., meeting ≥75% of estimated requirements) were significantly higher post intervention.

Factors associated with patients’ energy and protein intakes in proportion to estimated requirements are detailed in Table 3. There were no significant associations between the proportion of energy or protein requirements met with age, gender, diagnosis, or requiring feeding assistance. The main reasons for eating <50% of a meal were similar pre and post intervention, and included poor appetite, feeling too ill, experiencing nutrition impacting symptoms, being asleep, or disliking the meal.

Being prescribed oral nutrition support was associated with meeting a higher proportion of both EER and EPR (*p* < 0.05) and being seen by a dietitian was associated with meeting a higher proportion of EER (*p* < 0.05). Patients met a higher proportion of their estimated requirements when prescribed the ‘Full + Hot BF’ diet, as shown in Table 3.

There were no differences (*p* > 0.05) between groups (i.e., pre and post intervention) in terms of the number of patients who were seen by a dietitian (26%), received oral nutrition support (18%), required feeding assistance (21%), or ate < 50% of their meal (24%). The number of patients with the nutrition risk screening completed was greater pre (72%) than post (48%) intervention (*p* < 0.05).

### 3.3. Mealtime Observations

A total of 823 individual meals were observed over the study period; 423 pre and 400 post intervention. The total number of mealtime interruptions was higher after implementation of the intervention (111/423 meals pre and 150/400 meals post intervention, *p* < 0.001); however, the types of interruptions varied pre to post intervention. Table 4 describes types of mealtimes interruptions pre and post intervention.

The most common interruptions were nursing procedures, which included observations, handovers, bedmaking, toileting, and hygiene. Breakfast was the most interrupted meal (45% of breakfast meals interrupted), with lunch and dinner interrupted 34% and 11% of the time, respectively. The number of patients ready to receive their meal tray (i.e., positioned correctly at the time of meal tray delivery) increased from 76% to 84% after the intervention (*p* < 0.05).

## 4. Discussion

This study demonstrated significant improvements in patients’ protein intakes and in the number of patients eating adequately (in proportion to estimated requirements) for energy and protein, after implementation of a multifaceted, complex intervention targeting improved nutrition care on an acute medical ward. These improvements likely resulted from changes in nutrition care and practices at the individual patient, ward, and organisational levels.

Improvements in nutritional intakes are most likely to be due to the organisational change in introducing the ‘Full + Hot BF’ diet code, which gave patients the option to order more food; hence gave them access to more energy and protein. Previous studies [1,27] and the baseline data in this study have shown that patients consume the highest proportion of their meal at breakfast (compared to lunch and dinner). This may be due to patients having a better appetite and less nutrition impacting symptoms after an overnight fast, as well as the breakfast meal containing more familiar foods than meals served later in the day. Our baseline qualitative data from patient and family interviews support this. However, despite breakfast being the best consumed meal of the day, our pre-intervention data found that the breakfast meal contained the least energy and protein. Maximising intake at breakfast was seen by all key stakeholders as an appropriate strategy to improve patient intakes. Previous research has shown that regardless of the energy and protein content of a particular meal, patients still eat the same proportion or volume of the meal [28]. Hence, providing more energy and protein dense foods at this meal is likely to increase patients’ overall energy and protein intakes.

Another strategy to improve intakes among patients was changing the timing of breakfast delivery. In our baseline observations, the breakfast meal was being delivered 8:00–9:00 a.m. and clashing with medical and allied health ward rounds, resulting in frequent interruptions. Our study team and clinical stakeholders worked together with the hospital foodservice department to change breakfast delivery to 7:00–7:30 a.m. Interestingly, while there were fewer medical and allied health interruptions, there were more overall mealtime interruptions observed post-intervention than pre-intervention. The majority of interruptions occurred at the breakfast meal and most were nursing interruptions. This is probably because nursing activities (such as bed changing and hygiene) occur first thing in the morning and medication rounds begin at 7:00 a.m. Our findings are consistent with previous studies of protected mealtime interventions that either resulted in no change [11] or an increase [29] in mealtime interruptions post intervention. However, despite the increase in nursing interrputions in our study, patients’ nutritional intakes were still improved. It is possible that when nurses ‘interrupted’ a patient during their meal, they provided some kind of feeding setup or assistance, as this was encouraged in the nurse training aspect of the intervention. One study showed improvements in patient intakes with both (a) a protected mealtime and multidisciplinary staff education intervention; and (b) a feeding assistance intervention [30]. However, given most other studies using protected mealtime or feeding assistance interventions have found no effect on patients’ energy and protein intakes [11,29,31], the findings in this study are more likely to be related to the addition of a hot breakfast.

Other factors that may have contributed to improved nutrition intake among patients were increased awareness of nutrition among staff groups, which may have influenced their practice; and a change in ward culture regarding nutrition. These were achieved through the intervention itself (i.e., with multidisciplinary staff training) and also through the use of an integrated KT approach. This approach focuses on ‘co-development’ of interventions with end-users, so that interventions are more relevant, feasible, acceptable, and sustainable [17]. The research team involved end-users at multiple levels, by: 1) including key stakeholders as investigators on the study team; 2) establishing and liaising with a Nutrition Reference Group, which contained stakesholders representing staff from multiple disciplines as well as consumers (patients); and 3) having frequent, informal meetings with ward staff to disseminate results of each study phase and obtain ideas for intervention strategies. This ensured end-users were familiar with the study from the outset and had some ownership of the intervention. There were also several data collection periods (observational studies and interviews that involved staff and patients), which kept the project, and hence nutrition, at the forefront of people’s minds. Several other papers have highlighted the importance of creating a culture where nutrition is valued, of training multidisciplinary staff for specific roles related to nutrition care, and using a multi-level approach to improving nutrition that includes the organisation, staff, and patients and families as part of the solution [32,33,34].

This study has several limitations. Firstly, it was a pre/post design and not a randomised design, so it is not possible to guarantee the changes in patients’ nutrition intake was a direct result of the intervention. However, this was an integrated KT study, of pragmatic design, co-led by ward staff, so an RCT design is not possible. Staffs’ practice was changing slowly as a result of the integrated KT approach, even before the intervention was implemented (but after the baseline data collection period). There are coherent difficulties with ‘testing’ components of complex interventions and it is difficult to tease out which components of a complex intervention worked [20]. Process evaluations may help to understand what worked for whom under what conditions [35]; hence, we have conducted a process evaluation to give further insight into these findings (which will be reported separately). Secondly, this was a practical and pragmatic study, so we had to work within the limits of the hospital’s foodservice system and the ward’s resources and staffing when developing intervention strategies. However, on the positive side, this means the intervention was feasible and sustainable in practice. Some strengths of the study include the strong theory and evidence underpinning the intervention, as well as its development being guided by key frameworks [17,20]. This ensured intervention strategies were based on actual barriers and facilitators idenitified within the local context and developed in conjunction with key stakeholders. The rigorous collection of dietary intake data was another strength of the study, with patients being physically observed for their total period of daily dietary intake. This is a valid and reliable method of collecting intake data [24]; however, it is not always used in research due to it being resource-intensive.

## 5. Conclusions

This study found an improvement in patients’ nutritional intakes after the implementation of a complex, tailored intervention targeting nutritional practices on an acute medical ward. The success of the intervention was probably multi-factorial, due to the multiple intervention strategies and the integrated KT approach used. Future research requires a continued focus on pragmatic, context-specific interventions for improving nutrition among hospitalised patients. While this study shows promise, there is still progress to be made. For example, while the number of patients in this study who ate adequately (i.e., met >75% of their requirements) doubled post-intervention, this number only reached around 45%, highlighting the difficulty in patients reaching optimal nutrition in this setting. Nutrition interventions involving key stakeholders and end-users throughout the research process, that assess the problem within the local context and allow co-development of intervention strategies with these key people are more likely to be more acceptable, effective, and sustainable in practice.

## Figures and Tables

**Table 1 nutrients-11-01417-t001:** Patient demographics.

Demographic	Pre (*n* = 116)	Post (*n* = 91)	*p*-Value
Gender (female), *n* (%)	69 (59.5%)	48 (52.7%)	0.204 ^a^
Age (years), mean ± SD	73 ± 17	70 ± 17	0.357 ^b^
LOS, median (IQR)	3.5 (3.0–3.5)	4.0 (3.0–5.0)	0.265 ^c^
BMI, mean ± SD	26 ± 7	25 ± 7	0.376 ^b^

^a^ Chi-square test; ^b^ Independent samples *t*-test; ^c^ Mann–Whitney U test.

**Table 2 nutrients-11-01417-t002:** Patients’ energy and protein intakes and requirements pre and post implementation.

Nutrition Variable	Pre (*n* = 66)	Post (*n* = 61)	*p*-Value
Energy intake (kJ) mean ± SD	4818 ± 2179	5384 ± 1865	0.119 ^a^
Protein intake (g) mean ± SD	48 ± 24	57 ± 22	0.042 ^a,^*
Estimated Energy Requirement (kJ) mean ± SD	8025 ± 1748	7711 ± 1871	0.332 ^a^
Estimated Protein Requirement (g) mean ± SD	81 ± 15	75 ± 17	0.019 ^a,^*
Proportion Estimated Energy Requirement met (%) mean ± SD	60.1 ± 27.1	73.6 ± 32.0	0.015 ^a,^*
Proportion Estimated Protein Requirement met (%) mean ± SD	60.2 ± 30.5	80.0 ± 37.1	0.001 ^a,^*
*N* (%) patients who ate adequately – energy	13 (20%)	27 (44%)	0.003 ^b,^*
*N* (%) patients who ate adequately – protein	16 (24%)	28 (46%)	0.009 ^b,^*

^a^ Independent samples *t*-test; ^b^ Chi-square test; * *p* < 0.05 (statistically significant difference).

**Table 3 nutrients-11-01417-t003:** Proportion of energy and protein requirements met in terms of diet code.

Proportion of Requirements Met	Full Diet + Hot BF ^a^ (Post Only) *n* = 19	All Other Diets (Pre and Post) *n* = 108	All Other Diets (Post Only) *n* = 42
%EER met ^b^	80.5 ± 31.1	64.5 ± 29.5 (*p* = 0.032) ^c^	70.5 ± 32.3 (*p* = 0.261) ^c^
%EPR met ^b^	85.5 ± 41.5	66.9 ± 33.3 (*p* = 0.032) ^c^	77.5 ± 35.2 (*p* = 0.438) ^c^

^a^ New diet code implemented as part of intervention; ^b^ Presented as mean ± SD; ^c^ Compared to Full diet + hot BF (breakfast; post only).

**Table 4 nutrients-11-01417-t004:** Mealtimes interrupted pre and post intervention.

Interruption	Pre (423 meals)	Post (400 meals)	Total (823 meals)	*p*-Value
Nursing procedures *	66	85	151	0.023 *
Ward round	23	18	41	0.324
Medication round	14	58	72	<0.001 *
Patient taken off ward	5	3	8	0.394
Pathology	4	7	11	0.242

* May include multiple interruptions (i.e., one or more of the following: observations, handovers, bedmaking, toileting and hygiene).

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
