# Peer review of "Improving Nutrition Care, Delivery, and Intakes Among Hospitalised Patients: A Mixed Methods, Integrated Knowledge Translation Study"

_nutrients, 2019, doi:10.3390/nu11061417_

Round 1

Reviewer 1 Report

General comments

In this observational study, the investigators have used an integrated knowledge transfer approach to assess the impact of an intervention on the nutrition care, delivery and intake in acute case inpatients. This study, developed with different stakeholders is well designed and the report is well written. They should be commented for this work as such study involving caregivers with different backgrounds as well as patients is difficult to implement.

The authors give an accurate account of the strengths & weaknesses of this type of study in a discussion that is within the limit of the study. Hence I have but few minor comments.

Specific comments

The acronym “full + hot BF” should included when it is first cited in the text (Page 3, Line 110).

For the benefit of the readership that might not be entirely familiar, the authors should give a short description or definition of estimated energy requirements and of estimated protein requirements and how these were calculated (Page 4. Line144). This could be done as supplemental data.

The authors mention that patients’ mean proportion of estimated requirements met & the number of patients consuming adequate energy & protein were significantly higher post-intervention (Page 4, Lines170-171 & Page 5, Lines 172-173). To my understanding from table 2 this is true for the protein intake (P=0.042) but not for energy intake (P=0.119) or EER (P=0.332). I might be wrong but this requires clarification. The starting sentence of the discussion should be modified to reflect this (Page 6, Line 208) if needed.

Author Response

Review 1 comments

In this observational study, the investigators have used an integrated knowledge transfer approach to assess the impact of an intervention on the nutrition care, delivery and intake in acute case inpatients. This study, developed with different stakeholders is well designed and the report is well written. They should be commented for this work as such study involving caregivers with different backgrounds as well as patients is difficult to implement.The authors give an accurate account of the strengths & weaknesses of this type of study in a discussion that is within the limit of the study. Hence I have but few minor comments.

The authors thank the reviewer for their positive comments.

Specific comments:

The acronym “full + hot BF” should included when it is first cited in the text (Page 3, Line 110).

 The authors have added the acronym to Page 3, Line 110.

For the benefit of the readership that might not be entirely familiar, the authors should give a short description or definition of estimated energy requirements and of estimated protein requirements and how these were calculated (Page 4. Line144). This could be done as supplemental data.

 The authors have added the following: 

Institutional clinical practice guidelines were used to calculate individual, disease-specific estimated energy requirements (EER) and estimated protein requirements (EPR) [23]. These represent the predicted average daily dietary energy/protein intakes required to maintain energy balance and meet protein needs of individuals of a defined age, gender, weight and clinical condition, using static equations [23].

The authors mention that patients’ mean proportion of estimated requirements met & the number of patients consuming adequate energy & protein were significantly higher post-intervention (Page 4, Lines170-171 & Page 5, Lines 172-173). To my understanding from table 2 this is true for the protein intake (P=0.042) but not for energy intake (P=0.119) or EER (P=0.332). I might be wrong but this requires clarification. The starting sentence of the discussion should be modified to reflect this (Page 6, Line 208) if needed.

This has been amended to:

Patients’ mean proportion of estimated protein requirements met and the number of patients consuming adequate energy and protein (i.e. meeting ≥75% of estimated requirements) were significantly higher post intervention. (RESULTS)

AND:

This study demonstrated significant improvements in patients’ protein intakes and in the number of patients eating adequately (in proportion to estimated requirements) for energy and protein, after implementation of a multifaceted, complex intervention targeting improved nutrition care on an acute medical ward.  (DISCUSSION)

Reviewer 2 Report

This report concerns essentially a multimodality intervention to improve dietary intake. As such it is a difficult concept to study.

The approach however is clearly described and the report facilitates a reader to understand whether this approach might be interesting for application in own hospital (settings). 

Strenghts:

application (and study) of a multifaceted strategy

well structured approach

hospital-specific instrument

Weaknesses:

difficult to generalize the presented data

why analysis of only patients with full intake? Although a reletively high percentages, in a hospital setting, one is usually confronted with inadequate/insufficient (possibility) of intake. Is the current approach also working for those mostly in need for it? This is also reflected in the relatively high percentage of patients with adequate protein intake (table 2, 46%) which -at least in my gastroenterology ward- is hardly imaginable.

Do the authors have an explanation why being prescribed oral mutrition support would be helpful. Uually it is not, but apparently in this multifaceted setting it may be beneficial?

Author Response

This report concerns essentially a multimodality intervention to improve dietary intake. As such it is a difficult concept to study. The approach however is clearly described and the report facilitates a reader to understand whether this approach might be interesting for application in own hospital (settings). Strenghts: application (and study) of a multifaceted strategy; well structured approach hospital-specific instrument. Weaknesses: difficult to generalize the presented data.

The authors thank the reviewer for their positive comments, and acknowledge the limitations of generalisability as a single site study. 

why analysis of only patients with full intake? Although a reletively high percentages, in a hospital setting, one is usually confronted with inadequate/insufficient (possibility) of intake. Is the current approach also working for those mostly in need for it? This is also reflected in the relatively high percentage of patients with adequate protein intake (table 2, 46%) which -at least in my gastroenterology ward- is hardly imaginable.

By the term 'a full day's intake', we mean we observed all the patient's meals for that day - that is, we observed breakfast, lunch and dinner. It does not mean we only included patients who were eating well. We could only include patients for whom we had a complete snapshot of their daily intake for data analysis, otherwise the data would be missing and not representative.

Do the authors have an explanation why being prescribed oral mutrition support would be helpful. Uually it is not, but apparently in this multifaceted setting it may be beneficial?

This was an expected outcome, as literature suggests being prescribed oral nutrition support often results in higher nutritional intakes. Hence, this result was not surprising.